# Stability of SGD: Tightness Analysis and Improved Bounds

Yikai Zhang [*1]     Wenjia Zhang [*2]     Sammy Bald [*3]     Vamsi Pingali[4]     Chao Chen[5]     Mayank Goswami[3]

[1]Machine Learning Research., Morgan Stanley
[2]Computer Science Dept., Rutgers University
[3]Computer Science Dept., Queens College of CUNY
[4] Mathematics Dept., Indian Institue of Science
[5] Biomedical Informatics Dept., Stony Brook University

## Abstract

Stochastic Gradient Descent (SGD) based methods have been widely used for training large-scale machine learning models that also generalize well in practice. Several explanations have been offered for this generalization performance, a prominent one being algorithmic stability [Hardt et al., 2016]. However, there are no known examples of smooth loss functions for which the analysis can be shown to be tight. Furthermore, apart from properties of the loss function, data distribution has also been shown to be an important factor in generalization performance. This raises the question: is the stability analysis of [Hardt et al., 2016] tight for smooth functions, and if not, for what kind of loss functions and data distributions can the stability analysis be improved?

In this paper we first settle open questions regarding tightness of bounds in the data-independent setting: we show that for general datasets, the existing analysis for convex and strongly-convex loss functions is tight, but it can be improved for non-convex loss functions. Next, we give novel and improved data-dependent bounds: we show stability upper bounds for a large class of convex regularized loss functions, with *negligible regularization* parameters, and improve existing data-dependent bounds in the non-convex setting. We hope that our results will initiate further efforts to better understand the data-dependent setting under non-convex loss functions, leading to an improved understanding of the generalization abilities of deep networks.

## 1 INTRODUCTION

*Stochastic gradient descent* (SGD) has gained great popularity in solving machine learning optimization problems [Kingma and Ba, 2014, Johnson and Zhang, 2013]. SGD leverages the finite-sum structure of the objective function, avoids the expensive computation of exact gradients, and thus provides a feasible and efficient optimization solution in large-scale settings [Bottou, 2012]. The convergence and the optimality of SGD have been thoroughly studied [Ge et al., 2015, Rakhlin et al., 2012, Reddi et al., 2018, Zhou and Gu, 2019, Carmon et al., 2019a,b, Shamir and Zhang, 2013].

In recent years, new research questions have been raised regarding SGD's impact on a model's generalization power. The seminal work [Hardt et al., 2016] tackled the problem using the *algorithmic stability* of SGD, i.e., the progressive sensitivity of the trained model w.r.t. the replacement of a single (test) datum in the training set. They showed that the generalization error of an SGD-trained model is upper bounded by a uniform stability parameter $\varepsilon_{\text{stab}}$, and relate $\varepsilon_{\text{stab}}$ to the divergence of the two parameter vectors obtained by training on twin datasets.

This stability-based analysis of the generalization gap allows one to bypass classical model capacity theorems [Vapnik, 1998, Koltchinskii and Panchenko, 2000] or weight-based complexity theorems [Neyshabur et al., 2017, Bartlett et al., 2017, Arora et al., 2018]. This framework also provides theoretical insights into many phenomena observed in practice, e.g., the "train faster, generalize better" phenomenon, the power of regularization techniques such as weight decay [Krogh and Hertz, 1992], dropout [Srivastava et al., 2014], and gradient clipping. Other works have developed the stability notion with advanced analysis [Bassily et al., 2020, Feldman and Vondrak, 2019, Kuzborskij and Lampert, 2018, Lei and Ying, 2020b, Lei et al., 2021a, Lei and Ying, 2020a] and adapted it into more sophisticated settings such as Stochastic Gradient Langevin Dynamics and momentum SGD [Mou et al., 2018, Chaudhari et al., 2019, Chen et al.,

---

*These authors contributed equally to this work

*Accepted for the 38th Conference on Uncertainty in Artificial Intelligence* (UAI 2022).

2018, Li et al., 2020, Lei et al., 2021b].

Despite the promises of this stability-based analysis, it remains open whether the analysis in [Hardt et al., 2016] can be further improved to reveal the full potential of the stability method, either in general or for specific data-distributions.

**Our results:** We provide three kinds of results (see Table 1) that complement each other: a) tight lower bounds that show settings where stability analysis cannot be improved further for general datasets, b) weaker lower bounds that hint at a possible improvement, along with complementary improved upper bounds, also for general datasets and c) in settings where existing data-independent analysis cannot be improved, we derive improved data-dependent bounds. Below we summarize some of the existing open questions in this line of research, grouped according to properties of the loss function, along with our results addressing these problems.

## 1.1 CONVEX AND STRONGLY CONVEX LOSS

The following are the main results presented in [Hardt et al., 2016] for convex and strongly-convex loss functions (with certain Lipschitz and smoothness conditions), when optimized using SGD. Here $n$ denotes the size of the sample, $T$ the number of steps in SGD, and $\alpha_t$ the size of the SGD step in the $t$-th iteration.

1. For convex loss functions, the stability is upper bounded by $\sum_{i=1}^{T} \alpha_t/n$. The smaller the number of iterations $T$ is, the lower this upper bound. Hence "train faster, generalize better".

2. In practice, one often uses constant step size: $\alpha_t = \alpha$. For convex loss functions the upper bound would then scale linearly in the number of iterations $T$, which seems to be too pessimistic. [Hardt et al., 2016] show that by adding a $\frac{\mu}{2}\|w\|_2^2$ regularization term to the convex loss function, where $w$ is the vector of weights and $\mu \in \Theta(1)$ is a small constant, one gets much better stability upper bound for constant step size that does not depend on $T$, and is $O(1/n)$.

This gives rise to the following questions:

**Question 1:** Are the upper bounds of [Hardt et al., 2016] for convex and strongly-convex functions tight? That is, can one construct loss functions that satisfy the hypotheses and exhibit the claimed worst-case stability performance?

We remark that, to the best of our knowledge, the only construction available in the literature is [Bassily et al., 2020]. The authors analyze the stability of a loss function in order to derive lower bounds, but unfortunately, the loss function is not smooth and therefore does not satisfy the hypothesis in [Hardt et al., 2016].

**Question 2:** How important is the regularization term in order to make the transition from convex to strongly-convex,

and therefore the improvement from an $O(T/n)$ upper bound to an $O(1/n)$ upper bound for constant step-size SGD?

We provide the following answers to the above questions:

**Result 1:** The answer to question 1 is yes, i.e., there exist smooth, convex and strongly-convex loss functions that achieve the worst-case stability upper bound, In Theorem 1, we construct a Huber function which is quadratic in a certain area and linear outside. Under certain restricted assumptions, we proved the tightness of upper bounds in [Hardt et al., 2016] for convex loss which strengthens the lower bound of [Bassily et al., 2020] for the non-smooth case. In Theorem 2, our construction shows the tightness of upper bounds in [Hardt et al., 2016] for strongly convex loss.

**Result 2: (Data-dependent bounds)** We answer question 2 by introducing Theorem 3. In Theorem 3, we derive an upper bound on the stability for linear model loss function that is independent of $T$ (the number of iterations), even when the weight $\mu$ of the regularization term is very small (of the order of $1/n^4$), as long as the data satisfies a natural condition related to the Second Moment. Sharing a similar spirit with [Kuzborskij and Lampert, 2018], our result suggests that the property of distribution plays an important role in generalization of SGD, and nice properties of the data can almost replace the need for regularization.

## 1.2 NON-CONVEX LOSS

[Hardt et al., 2016] also prove an upper bound for non-convex loss functions, and one wonders again whether the bound is tight. After only being able to prove a slightly weaker lower bound, we realized that this was because one can actually improve the analysis in [Hardt et al., 2016]!

**Result 3:** We provide matching lower ( Theorem 4) and upper bounds ( Theorem 5) on the stability of SGD for non-convex functions, that are tighter than the upper bound in [Hardt et al., 2016] for a wide and interesting range of values of $T$ (e.g., when $n < T < n^{10}$).

In the non-convex setting, the bounds in both [Hardt et al., 2016] and our Result 3 assume a decreasing step-size $\alpha_t \propto 1/t$ in SGD. However, in practice the constant step-size case is very important. Although it is not derived formally, the techniques in [Hardt et al., 2016] can be employed to show an *exponential* upper bound for non-convex loss functions minimized using SGD with constant-size step, raising the question of the existence of better analysis.

**Result 4:** Also by Theorem 7, we show that without any additional assumptions on either the loss function or the data distribution, improving on this analysis is hopeless by providing a lower bound that is exponential in $T$.

Data-dependent bounds: This naturally raises the question

Table 1: Current landscape of stability bounds. [H] indicates results in [Hardt et al., 2016], [K] indicates results in [Kuzborskij and Lampert, 2018] and * indicates results in this paper. $\beta$ is the smoothness parameter. $\zeta$ is a data-dependent constant defined in Lemma 5. $\widehat{\varepsilon}_{\text{stab}}$ is on-average stability defined in Def 7. $a$, $b$ are small constants free of $T$ and $n$. We only keep $T$ and $n$ term in the bounds.

| SGD Step Size | Constant $\alpha_t = a/\beta$ | | $\alpha_t = a/(\beta t)$ | $\alpha_t = b/t$ |
|---|---|---|---|---|
| *Loss function* | Strongly Convex | Convex | Non-Convex | Non-Convex with $\widehat{\varepsilon}_{\text{stab}}$ |
| *Upper Bound* | $O(\frac{1}{n})$ [H] | $O(T/n)$ [H] | $O\left(T^{\frac{a}{1+a}}/n\right)$ [H] $\quad$ $O\left(T^a/n^{1+a}\right)$* | $O\left(T^{\frac{\zeta b}{1+\zeta b}}/n\right)$[K] $\quad$ $O(T^{\zeta b}/n^{1+\zeta b})$* |
| *Lower Bound* | $\Omega(\frac{1}{n})$* | $\Omega(T/n)$* | $\Omega(T^a/n^{1+a})$* | Open |

of deriving data-dependent bounds on stability in the non-convex setting. The work in [Kuzborskij and Lampert, 2018] took the first step in this direction by analyzing SGD using concept of "average stability" from [Bousquet and Elisseeff, 2002, Shalev-Shwartz et al., 2010], and deriving upper bounds on it. Finally, we show:

**Result 5:** The improved analysis for uniform stability of SGD on non-convex and smooth loss functions can also be applied to improve on the result in [Kuzborskij and Lampert, 2018] and obtain a tighter bound for the average stability of SGD. We present Theorem 6 as the data-dependent version of Theorem 5.

In summary, we essentially close the open questions of tightness in data-independent settings for all three classes of functions, and improve upper bounds in the data-dependent setting. We hope that our results will initiate further efforts to better understand the data-dependent setting under non-convex loss functions and analyze the conditions under which one can expect better upper bounds on stability and generalization of SGD.

## 2 RELATED WORKS

The stability framework suggests that a stable machine learning algorithm results in models with good generalization performance [Kearns and Ron, 1999, Bousquet and Elisseeff, 2002, Elisseeff et al., 2005, Shalev-Shwartz et al., 2010, Devroye and Wagner, 1979a,b, Rogers and Wagner, 1978, Bousquet and Elisseeff, 2002]. It serves as a mechanism for provable learnability when uniform convergence fails [Shalev-Shwartz et al., 2010, Nagarajan and Kolter, 2019]. The concept of uniform stability was introduced in order to derive high probability bounds on the generalization error [Bousquet and Elisseeff, 2002]. Uniform stability describes the worst-case change in the loss of a model trained on an algorithm when a single data point in the dataset is replaced. In [Hardt et al., 2016], a uniform stability analysis for *iterative algorithms* is proposed to analyze SGD, generalizing the one-shot version in [Bousquet and Elisseeff, 2002]. Algorithmic uniform stability is widely used in analyzing the generalization performance of SGD [Mou et al., 2018,

Feldman and Vondrak, 2019, Chen et al., 2018]. The worst-case leave-one-out type bounds also closely connect uniform stability with *differential private learning* [Feldman et al., 2018, 2020, Dwork et al., 2006, Wu et al., 2017], where the uniform stability can lead to provable privacy guarantees. Beside uniform stability, [Liu et al., 2017] proposed *argument stability* to capture stability of selected hypothesis function space.

While the upper bounds of algorithmic stability of SGD have been extensively studied, the tightness of those bounds remains open. In addition to uniform stability, an *average stability* of the SGD is studied in [Kuzborskij and Lampert, 2018] where the authors provide *data-dependent* upper bounds on stability[1]. Our analysis framework for deriving improved bounds in [Hardt et al., 2016] can also be applied to improve the data-dependent stability results in [Kuzborskij and Lampert, 2018].

In [Bassily et al., 2020], a lower bound on the stability of SGD for nonsmooth convex losses is proposed. The lower bound is designed to illustrate the tightness of the stability analysis *without* smoothness assumptions. In this work, we report for the first time lower bounds on the uniform stability of SGD for smooth loss functions. Our tightness analysis suggests the necessity of additional assumptions for analyzing the generalization of SGD for deep learning.

## 3 PRELIMINARIES

In this section we introduce the notion of uniform stability and establish notation. We first introduce the quantities *empirical risk*, *population risk*, and *generalization gap*. Given an unknown distribution $\mathcal{D}$ on labeled sample space $Z = X \times \mathbb{R}$, let $S = \{z_1, ..., z_n\}$ denote a set of $n$ samples $z_i = (x_i, y_i)$ drawn i.i.d. from $\mathcal{D}$. Let $w \in \mathbb{R}^d$ be the parameter(s) of a model that predicts $y$ given $x$, and let $f$ be a loss function where $f(w; z)$ denotes the loss of the model with

---

[1]While it is an interesting open problem to get data-dependent lower bounds by lower bounding the average stability, we construct lower bounds on the worst-case stability. Thus our lower bounds are general and not data-dependent.

parameter(s) $w$ on sample $z$. Let $f(w; S)$ denote the *empirical risk* $f(w; S) = E_{z \sim S}[f(w; z)] = \frac{1}{n} \sum_{i=1}^{n} f(w; z_i)$ with corresponding *population risk* $E_{z \sim \mathcal{D}}[f(w; z)]$. The *generalization error* of the model with parameter(s) $w$ is defined as the difference between the empirical and population risks:

$$|E_{z \sim \mathcal{D}}[f(w; z)] - E_{z \sim S}[f(w; z)]|.$$

Next we introduce *stochastic gradient descent* (SGD). We follow the setting of [Hardt et al., 2016]: starting with initialization $w_0 \in \mathbb{R}^d$, an SGD update step takes the form

$$w_{t+1} = w_t - \alpha_t \nabla_w f(w; z_{i_t})$$

where $i_t$ is drawn from $[n] = \{1, 2, \cdots, n\}$ uniformly and independently in each round. Let $\mathcal{W}$ be a convex and compact set to be optimized over. For projected SGD we let

$$w_{t+1} = \Pi_{w \in \mathcal{W}}\left(w_t - \alpha_t \nabla_w f(w; z_{i_t})\right)$$

where $\Pi_{\mathcal{W}}(v) = \arg\min_{w \in \mathcal{W}} \|w - v\|$.

The analysis of SGD requires the following crucial properties of the loss function $f(\cdot, z)$ at any fixed point $z$, viewed solely as a function of the parameter(s) $w$:

**Definition 1** ($L$-Lipschitz)**.** *A function $f(w)$ is $L$-Lipschitz if $\forall u, v \in \mathbb{R}^d$: $|f(u) - f(v)| \leq L\|u - v\|$.*

**Definition 2** ($\beta$-smooth)**.** *A function $f(w)$ is $\beta$-smooth if $\forall u, v \in \mathbb{R}^d$: $|\nabla f(u) - \nabla f(v)| \leq \beta \|u - v\|$.*

**Definition 3** ($\gamma$-strongly-convex)**.** *A function $f(w)$ is $\gamma$-strongly-convex if $\forall u, v \in \mathbb{R}^d$:*

$$f(u) > f(v) + \nabla f(v)^\top [u - v] + \frac{\gamma}{2} \|u - v\|^2.$$

**Definition 4** ($\rho$-Lipschitz Hessian)**.** *A loss function $f$ has a $\rho$-Lispchitz Hessian if $\forall u, v \in \mathbb{R}^d$, $\|\nabla^2 f(u) - \nabla^2 f(v)\| \leq \rho \|u - v\|$.*

**Algorithmic Stability:** Next we define the key concept of *algorithmic stability*, which was introduced by [Bousquet and Elisseeff, 2002] and adopted by [Hardt et al., 2016]. Informally, an algorithm is *stable* if its output only varies slightly when we change a single sample in the input dataset. When this stability is *uniform* over all datasets differing at a single point, this leads to an upper bound on the generalization gap. We now flesh this out more formally.

**Definition 5.** *Two sets of samples $S, S'$ are twin datasets if they differ at a single entry, i.e., $S = \{z_1, \dots z_i, \dots, z_n\}$ and $S' = \{z_1, \dots, z_i', \dots, z_n\}$.*

Now, let $\mathcal{A}$ be a (possibly randomized) algorithm which is parameterized by a sample $S$ of $n$ datapoints as $\mathcal{A}(S)$.

**Definition 6.** *(Stability) Define the algorithmic stability parameter $\varepsilon_{stab}(\mathcal{A}, n)$ as*

$$\inf\{\varepsilon : \sup_{z, S, S'} \mathbb{E}_{\mathcal{A}} |f(\mathcal{A}(S); z) - f(\mathcal{A}(S'); z)| \leq \varepsilon\}.$$

The expectation $\mathbb{E}_{\mathcal{A}}$ factors in the possible randomness of $\mathcal{A}$. For such an algorithm, one can define its expected generalization error as

$$GE(\mathcal{A}, n) := \mathbb{E}_{S, \mathcal{A}}[\underset{z \sim \mathcal{D}}{E}[f(\mathcal{A}(S); z)] - \underset{z \sim S}{E}[f(\mathcal{A}(S'); z)]].$$

We also define a data-dependent stability which is an average stability that was introduced by [Rakhlin et al., 2005, Shalev-Shwartz et al., 2010] and was applied for analyzing algorithmic stability of SGD by [Kuzborskij and Lampert, 2018].

**Definition 7** (On-average stability)**.** *Let $\mathcal{D}$ be the data distribution and $w_0$ be the initialized weight. A randomized algorithm $\mathcal{A}$ is $\widehat{\varepsilon}_{stab}(\mathcal{D}, w_0)$-on-average stable if*

$$\mathbb{E}_{S, S'} \mathbb{E}_{\mathcal{A}}[f(\mathcal{A}_S; z) - f(\mathcal{A}_{S'}; z)] \leq \widehat{\varepsilon}_{stab}(\mathcal{D}, w_0),$$

*where $S \overset{iid}{\sim} \mathcal{D}^m$ and $S'$ is its copy with $i$-th example replaced by $z \overset{iid}{\sim} \mathcal{D}$.*

Throughout this paper, we will write $\varepsilon_{\text{stab}}$ and $\widehat{\varepsilon}_{\text{stab}}$ omitting dependencies that are clear in context.

**Stability and generalization:** It was proved in [Hardt et al., 2016] that $GE(\mathcal{A}, n) \leq \varepsilon_{\text{stab}}(\mathcal{A}, n)$. Furthermore, the authors observed that an $L$-Lipschitz condition on the loss function $f$ enforces a uniform upper bound: $\sup_{z \in Z} |f(w; z) - f(w'; z)| \leq L\|w - w'\|$. This implies that for a Lipschitz loss, the algorithmic stability $\varepsilon_{\text{stab}}(\mathcal{A}, n)$ (and hence the generalization error $GE(\mathcal{A}, n)$) can be bounded by obtaining bounds on $\|w - w'\|$. And in [Kuzborskij and Lampert, 2018] they have similar results in the notion of on-average stability.

Let $w_t$ and $w_t'$ be the parameters obtained by running SGD on twin datasets $S, S'$ respectively for $t$ iterations. The *divergence quantity* is defined as $\delta_t := \mathbb{E}_{\mathcal{A}} \|w_t - w_t'\|$. While [Hardt et al., 2016] reports upper bounds on $\delta_t$ for different loss functions, e.g., convex and non-convex loss functions, we investigate the tightness of those bounds.

## 4   MAIN RESULTS

In this section, we report our main results. We first consider the convex case with constant step size, where we prove 1) that the existing bounds in [Hardt et al., 2016] are tight, and 2) for linear models, we report a data-dependent analysis to show that $\varepsilon_{\text{stab}}$ does not increase with $t$. Then we move on to the non-convex case, where a) for decreasing step size we report a lower bound suggests that within a wide range of $T$,

existing bound in [Hardt et al., 2016] is not tight. We prove a tighter upper bound which matches our lower bound thus, and b) for constant step size we give loss functions whose divergence $\delta_t$ increases exponentially with $t$.

## 4.1 CONVEX CASE

In this section we analyze the stability of SGD when the loss function is convex and smooth. We begin with a construction which shows that Theorem 3.8 in [Hardt et al., 2016] is tight. Our lower bound analysis will require the quadratic function

$$f(w; z) = \frac{1}{2} w^\top A w - y x^\top w, \quad (1)$$

where $A$ is a $d \times d$ matrix. In the construction of lower bounds, we carefully choose $A$ and $S$ so that the single data point replaced in the twin data set will cause the instability of SGD. In particular, we will choose $A$ to be a PSD matrix in the convex case in the construction of the lower bound and choose $A$ to be an indefinite matrix with some strictly negative eigenvalues in the non-convex case. We first begin with the following lemma which describes how $\|w_t - w_t'\|$ behaves for functions defined in Equation 1.

**Lemma 1** (Dynamics of divergence). *Let $f(w; x) = \frac{1}{2} w^\top A w - y x$. Suppose $[x_i - x_i']/\|x_i - x_i'\|$ is an eigenvector of $A$, i.e., $A[x_i - x_i'] = \lambda_{xx'}[x_i - x_i']$. Let $\Delta_t$ be $w_t - w_t'$, $\alpha_t \le \lambda_{xx'}$ be the step size of SGD and $\Delta_0 = 0$. If one runs SGD on $f(w, S)$ and $f(w, S')$ where $S, S'$ are twin datasets and $x_i'^\top x_j = 0, x_i^\top x_j = 0, \ \forall j \ne i$, then the dynamics of $\Delta_t$ are given by*

$$\mathbb{E}_{\mathcal{A}} \|\Delta_{t+1}\| = (1 - \alpha_t \lambda_{xx'}) \mathbb{E}_{\mathcal{A}} \|\Delta_t\| + \frac{\alpha_t}{n} \|x_i - x_i'\|. \quad (2)$$

**Remark 1.** *In this work, we assume that the different entry data $x_i, x_i'$ are orthogonal to all other samples. Such a restrictive setting serves as a corner case to prove the tightness, suggesting the necessity of additional assumptions to improve the upper bound. Indeed in Theorem 5, we introduce more realistic assumptions to avoid such corner cases, and the upper bound can be improved accordingly.*

The next lemma recursively applies Lemma 1. We will carefully chose $\lambda_{xx'}$ in the following lemma for lower bound constructions in the convex and non-convex cases.

**Lemma 2** (Lower bound on divergence). *Let $f(w; x) = \frac{1}{2} w^\top A w - y x$. Suppose $[x_i - x_i']/\|x_i - x_i'\|$ is an eigenvector of $A$ where $A[x_i - x_i'] = \lambda_{xx'}[x_i - x_i']$. Let $\Delta_t$ be $w_t - w_t'$, $\alpha_t \le \lambda_{xx'}$ be the step size of SGD and $\Delta_0 = 0$. If one runs SGD on $f(w, S)$ and $f(w, S')$ where $S, S'$ are twin datasets and $x_i'^\top x_j = 0, x_i^\top x_j = 0, \ \forall j \ne i$, then we have*

$$\mathbb{E}_{\mathcal{A}} \|\Delta_T\| \ge \frac{\|x_i - x_i'\|}{n} \sum_{t=1}^{T-1} \prod_{\tau=t+1}^{T-1} \alpha_t (1 - \alpha_\tau \lambda_{xx'}).$$

Now we can present our tightness results. We begin with the convex case. The main idea of the construction is to leverage Equation 1 with specially designed $A$ and $S, S'$ to ensure that $\mathbb{E}_{\mathcal{A}} \|w_T - w_T'\|$ will diverge. However, quadratic function in general does not $L$-Lipschitz condition, which does not match the assumption used to derive upper bound in Hardt et al. [2016]. To obtain the $L$-Lipschitz condition, we trim $f(w; S)$ to mimic the Huber loss function [Huber, 1992] so that the smoothness is maintained for the piecewise function.

**Theorem 1** (Lower bound for convex losses). *Let $w_t, w_t'$ be the outputs of SGD on twin datasets $S, S'$ respectively. Let $\Delta_t = w_t - w_t'$ and $\alpha_t$ be the step size of SGD. There exists a function $f$ which is convex, $\beta$-smooth, and $L$-Lipschitz, and twin datasets $S, S'$ such that*

$$\varepsilon_{stab} \ge \frac{L}{2n} \sum_{t=1}^{T} \alpha_t. \quad (3)$$

The convex upper bound in Theorem 3.8 of [Hardt et al., 2016] states that $\mathbb{E}_{\mathcal{A}} \|\Delta_T\| \le \sum_{i=1}^{T} \frac{\alpha_t L}{n}$, which implies that the divergence increases throughout training. The lower bound in Theorem 1 suggests the tightness of the upper bound. However, in practice, this is not commonly observed; the generalization performance does not deteriorate as the number of training iterations increases. Under the $\gamma$-strongly-convex loss function condition, [Hardt et al., 2016] provides an $O(\frac{1}{n})$ uniform stability bound, which fits better with empirical observations on classical convex losses. In the next theorem, we show the tightness of the $O(\frac{1}{n})$ bound for strongly-convex losses.

**Theorem 2** (Lower bound for strongly-convex losses). *Let $w_t, w_t'$ be the outputs of SGD on twin datasets $S, S'$ respectively, $\Delta_t$ be $w_t - w_t'$ and $\alpha = \frac{1}{2\beta}$ be the step size of SGD. There exists a function $f$ which is $\gamma$-strongly-convex and $\beta$-smooth, and twin datasets $S, S'$ such that the divergence and stability of the two SGD outputs satisfies*

$$\varepsilon_{stab} \ge \frac{1}{16\gamma n}. \quad (4)$$

Theorem 2 provides evidence for the tightness of the $O(\frac{1}{n})$ stability bound on SGD. To obtain such stability, the loss function must satisfy $\nabla_w^2 f(w; z) > \gamma I_d$ with $\gamma = \Omega(1)$. In general this does not hold, e.g., the Hessian of an individual linear regression loss term is $x_j x_j^\top$ which is not strongly-convex. In practice one can incorporate a strongly-convex regularizer to impose strong convexity, often resulting in improved generalization performance in practice [Shalev-Shwartz et al., 2010, Bousquet and Elisseeff, 2002]. However, an $O(1)$ regularization term will bias the loss function away from achieving sufficiently low empirical risk. This motivates us to investigate a weaker condition than strong convexity which still can enforce an $O\left(\frac{1}{n}\right)$ stability, without substantially biasing the loss function.

In the remainder of this section, we restrict ourselves to a family of linear model loss functions and show that the $O(\frac{1}{n})$ stability results can be obtained under the framework of average stability. The results of Theorem 3 have a dependence on a property of the distribution, and are thus distribution-dependent. We begin with the definition of a $\xi$-bounded Second Moment. Essentially, a bounded Second Moment dataset requires an average linear dependence of $Span\{x_1, ..., x_n\}$. Recall that the $i$-th sample is of the form $z_i = (x_i, y_i)$.

**Definition 8.** *A set $S = \{(x_1, y_1), ..., (x_n, y_n)\}$ is defined to have $\xi_S$- bounded Second Moment if $\forall v \in Span\{x_1, ..., x_n\}$*

$$v^\top (\frac{1}{n} \sum_{i=1}^n x_i x_i^\top) v \geq \xi_S v^\top v.$$

*A distribution $\mathcal{D}$ has a $(\xi, n, \mu)$-inversely bounded Second Moment if there exists a constant $\xi > 0$ such that*

$$\mathbb{E}_{S \sim \mathcal{D}^n} \left[ \frac{1}{\xi_S + \mu} \right] \leq \frac{1}{\xi + \mu}.$$

**Remark 2.** *The value of $\xi_S$ is always lower bounded by the minimum nonzero eigenvalue of $\frac{1}{n} \sum_j x_j x_j^\top$ which is the empirical second moment of data with size $n$.*

**Proposition 1** (Example of distribution with inversely bounded Second Moment). *Let $\mathbb{E}_{x \sim \mathcal{D}}[xx^\top] = \Sigma$ and $\xi$ be the minimum non-zero eigenvalue of $\Sigma$. Suppose $S = \{(x_1, y_1), ..., (x_n, y_n)\}$ is sampled from $\mathcal{D}$ with the $x \in \mathbb{R}^d$ with $\|x\| \leq 1$. Then, there exists universal constant $C, c$ so that if $n \geq max\{\frac{4C^2 d}{\xi^2}, \frac{512}{c\xi^2} \log(\frac{1}{\xi})\}$, $\mathcal{D}$ has a $(\frac{\xi}{3}, n, \mu)$-inversely bounded Second Moment if $\mu \geq \frac{1}{n^4}$.*

In our next theorem, we leverage the inversely bounded Second Moment property to prove a non-accumulated on-average stability bound for SGD on *linear models* with a regularized loss function. We characterize a linear model by rewriting the loss function $f(w; z)$ in terms of $f_y(w^\top x)$ where $f_y(\cdot)$ is a scalar function depending only on the inner product of the model parameter $w$ and the input feature $x$.

**Theorem 3** (Data-dependent stability of SGD with inversely bounded Second Moment). *Suppose a loss function $f(w, z)$ is of the form*

$$f(w, S) = \frac{1}{n} \sum_{j=1}^n f_{y_j}(w^\top x_j) + \frac{\mu}{2} w^\top w \; ; w \in \mathcal{W}$$

*where $f_y(w^\top x)$ satisfies (1) $|f_y'(\cdot)| \leq L$ , (2) $0 < \gamma \leq f_y''(\cdot) \leq \beta$, (3) $S, S'$ are sampled from $\mathcal{D}$ with $\xi$ be the minimum nonzero eigenvalue of $\mathbb{E}_{x \sim \mathcal{D}}[xx^\top]$ and a uniformly bounded support $\mathcal{X} : \|x\| \leq 1, \mathcal{X} \subset \mathbb{R}^d$ and 4) $\mu \geq \frac{\gamma}{n^4}$. Let $\mathcal{W}$ be a convex and compact set, $w_t$ and $w_t'$ be the outputs of SGD on $S$ and $S'$ after $t$ steps, respectively. Let*

*the divergence $\Delta_t := w_t - w_t'$ and $\alpha \leq \frac{\mu}{2\beta^2}$ be the step size of SGD. There exists universal constant $C, c$ so that if $n \geq max\{\frac{4C^2 d}{\xi^2}, \frac{512}{c\xi^2} \log(\frac{1}{\xi})\}$, then*

$$\mathbb{E}_S \mathbb{E}_{\mathcal{A}} \|\Delta_T\| \leq \frac{12L}{\xi \gamma n}, \; and \; \widehat{\varepsilon}_{stab}(\mathcal{D}) \leq \frac{16L^2}{\xi \gamma n}.$$

**Remark 3.** *The inversely bounded Second Moment condition allows SGD to maintain an average stability guarantee for a family of widely used models with a negligible regularizer and large sample size. The theorem suggests that if the dataset $S$ is sampled from a 'good' distribution, one can obtain an advanced generalization property which mainly depends on the distribution. The theorem also justifies the common choice of small values for the weight in the $L_2$-regularizer (also known as weight decay) when training ridge regression type models. Note that the term $\frac{\mu}{2} w^\top w$ makes the loss function strongly convex, and a $O\left(\frac{1}{n}\right)$ is established with $\mu = O(1)$ in [Hardt et al., 2016]. The major difference of Theorem 3 is that the weight of the $\ell_2$ penalty $\mu$ is $O\left(\frac{1}{n^4}\right)$ for uniformly bounded $x$. A small value of $\mu$ will not bias the original loss function thus allow the SGD to sufficiently minimize the empirical risk. In stead of leveraging the $\ell_2$ penalty, the stability of SGD is obtained upon the 'nice' property of the distribution.*

**Example: Linear regression.** Linear regression minimizes the quadratic loss on $w$: $f(w, S) = \frac{1}{2n} \sum_{x_j \in S} (x_j^\top w - y_j)^2, w \in \mathcal{W}$, where $\mathcal{W}$ is a convex compact set that contains the origin and has bounded radius $R$. The Hessian of an individual linear regression loss term is $x_j x_j^\top$ which is *not strongly-convex*. However, one can rewrite the loss function as $f_y(w^\top x)$ where $f_y''(\cdot) = 1$. Next we present certain conditions that are sufficient to make $|f'(\cdot)| \leq L$. We assume $\|x_i\| = 1, y_i \in [-1, 1], \forall i \in [n]$. Let $\Pi_{\mathcal{W}}(v) = \arg\min_{w \in \mathcal{W}} \|w - v\|$. Note that SGD updates as $w_{t+1} = \Pi_{w \in \mathcal{W}} \left( w_t - \alpha_t(x_j^\top w_t - y_j) x_j \right)$. One can show that $\sup_{w \in \mathcal{W}} \sup_{x, y \in S} f_y'(w^\top x) \leq R + 1$.

## 4.2 NON-CONVEX CASE

In this section, we construct a non-convex loss function to analyze the tightness of the divergence bound in [Hardt et al., 2016]. We first focus on the case where SGD applies a step size that *decreases with $t$*. Define a *hitting time* to be the time $t$ that satisfies $w_{t-1} - w_{t-1}' = 0$ and $w_t - w_t' \neq 0$. We first fix a hitting time $t_0$ and prove Lemma 3.

**Lemma 3** (Divergence of non-convex loss function). *There exists a function $f$ which is non-convex and $\beta$-smooth, twin datasets $S, S'$ and constant $a > 0$ such that the following holds: if SGD is run using step size $\alpha_t = \frac{a}{0.99\beta t}$ for $1 \leq t < T$, and $w_t, w_t'$ are the outputs of SGD on $S$ and $S'$, respectively, and $\Delta_t = w_t - w_t'$, then $\forall 1 \leq t_0 \leq T, \quad \mathbb{E}_{\mathcal{A}} \left[ \|\Delta_T\| | \Delta_{t_0} \neq 0 \right] \geq \frac{1}{2n} \left( \frac{T}{t_0} \right)^a$.*

The following theorem follows from Lemma 3 by optimizing over $t_0$. The choice of hitting time $t_0$ plays an important role in the analysis, which is also illustrated in the "burn-in Lemma" 3.11 in [Hardt et al., 2016].

**Theorem 4** (Lower bound for non-convex loss functions). *Let $w_t, w_t'$ be the outputs of SGD on twin datasets $S, S'$, and $\Delta_t = w_t - w_t'$. There exists a function $f$ which is non-convex and $\beta$-smooth, twin datasets $S, S'$ and constants $a < 0.1$ such that the divergence of SGD after $T > n$ rounds using constant step size $\alpha_t = \frac{a}{0.99\beta t}$ satisfies*

$$\varepsilon_{stab} \geq \frac{T^a}{6n^{1+a}}. \tag{5}$$

**Remark 4.** *In the above theorem, we require $\alpha_t = \frac{a}{0.99\beta t}$ with an extra constant factor $\frac{1}{0.99}$ to apply the inequality $1 + \frac{ax}{0.99} > e^{ax}$ with sufficiently small $a$. To remove the constant $1/0.99$ in the learning rate one need to avoid using the inequality $1 + x < e^x$ at the first place in deriving the upper bound. This can be done by a refined analysis for upper bound via setting learning rate $\alpha_t = \frac{e^{\frac{a}{t}} - 1}{\beta}$.*

**Remark 5.** *Note in [Hardt et al., 2016], an assumption is made on the non-convex loss function, namely that $f(u, z) \in (0, 1)$. In our lower bound construction, we do not have such an assumption thus our lower bound can not be directly compared with the upper bound in [Hardt et al., 2016]. The bound in [Hardt et al., 2016] is of the form $O\left(\frac{T^{\frac{a}{1+a}}}{n}\right)$, for $T^{\frac{a}{1+a}} \geq n$, our lower bound will exceed the upper bound in [Hardt et al., 2016]. However, such a gap implies that even with additional assumptions in [Hardt et al., 2016], the upper bound still may not be tight. The lower bound is derived by choosing the hitting time $t_0 < n$, i.e., the first time SGD picks the different entries $z, z'$ in the twin dataset before round $n$, suggesting additional space for improvement on the analysis. We investigate this gap and derive a tighter bound in the next theorem which improves on Theorem 3.12 in [Hardt et al., 2016].*

To prove a better upper bound for non-convex losses, we need the following lemma, which gives us the expectation of divergence for a given hitting time $t_k + 1$, which is the timestamp of SGD first selecting the $k$-th different sample.

**Lemma 4.** *[Hardt et al., 2016] Assume $f$ is $\beta$-smooth and $L$-Lipschitz. Let $w_t, w_t'$ be outputs of $SGD$ on twin datasets $S, S'$ respectively after $t$ iterations and let $\Delta_t = [w_t - w_t']$ and $\delta_t = \mathbb{E}\|\Delta_t\|$. Running SGD on $f(w; S)$ with step size $\alpha_t = \frac{a}{\beta t}$ satisfies the following conditions:*

- *The SGD update rule is a $(1 + \alpha_t\beta)$-expander and $2\alpha_t L$-bounded.*
- *$\mathbb{E}_{\mathcal{A}}[\|\Delta_t\| | \Delta_{t-1}] \leq (1 + \alpha_t\beta)\|\Delta_{t-1}\| + \frac{2\alpha_t L}{n}$.*
- *$\mathbb{E}_{\mathcal{A}}[\|\Delta_T\| | \Delta_{t_k} = 0] \leq \left(\frac{T}{t_k}\right)^a \frac{2L}{n}$.*

Lemma 4 bounds the case when the hitting time is equals to $t_k$. In the proof of Hardt et al. [2016] for non-convex stability upper bound, the $t_k$ is chose to be $T^{\frac{a}{1+a}}$. However, we observe that a choice of $t_k$ with additional care on the analysis leads to an improved upper bound. Therefore, we could obtain the upper bound for the stability of uniform sampling SGD as follows:

**Theorem 5** (Uniform sampling SGD). *Assume $f$ is $\beta$-smooth and $L$-Lipschitz. Running $T > n$ iterations of SGD on $f(w; S)$ with step size $\alpha_t = \frac{a}{\beta t}$, the stability of SGD satisfies*

$$\varepsilon_{stab} \leq \frac{16L^2 T^a}{n^{1+a}}.$$

We remark that the above analysis is for uniform sampling SGD, where the algorithm keeps sampling with replacement. We also derive a version of Theorem 5 which samples without replacement in the appendix, which also matches the lower bound. Dividing our bound by the bound in Theorem 3.12 of [Hardt et al., 2016], we obtain the ratio $\tilde{\Omega}\left(\frac{T^{\frac{a^2}{1+a}}}{n^a}\right)$. This factor is less than 1 (and so we improve the upper bound) exactly when $T^{\frac{a}{1+a}} \leq n$. Note that this is potentially a large range as $a$ is a small and positive constant.

In [Kuzborskij and Lampert, 2018], the data-dependent stability of SGD is analyzed, incorporating the dependence on the variance of SGD curvature and the loss of the initial parameter $w_0$ in analyzing the divergence of SGD. This framework has applications in transfer learning, as well as implications including optimistic generalization error. We observe that our analysis in Theorems 5 can be combined with the data-dependent framework, and we now report our data-dependent versions of Theorems 5.

The analysis requires the additional bounded variance assumption for SGD such that

$$\mathbb{E}_{S,z}\left[\|\nabla f(w_t; z) - \nabla\mathbb{E}_z(f(w_t; z))\|^2\right] \leq \sigma^2, \quad \forall t.$$

In the rest of this section we assume the variance of SGD satisfies this property.

We borrow the following lemma from [Kuzborskij and Lampert, 2018] which is a data-dependent version of Lemma 4.

**Lemma 5.** *[Kuzborskij and Lampert, 2018] Assume $f$ is $\beta$-smooth, $L$-Lipschitz, and has a $\rho$-Lipschitz Hessian. With $w_0$ the initial weight and $w_t, w_{t'}$ the outputs of SGD on twin datasets $S, S'$ respectively after $t$ iterations, let $\Delta_t = [w_t - w_{t'}]$. Running SGD on $f(w; S)$ with step size $\alpha_t = \frac{b}{t}$ where $b \leq \min\{\frac{2}{\beta}, \frac{1}{8\beta^2 \ln T^2}\}$ has the following properties:*

1. *The SGD update rule is a $(1 + \alpha_t\psi_t)$-expander and $\alpha_t L$-bounded. Here $\psi_t = \min\{\beta, \kappa_t\}$ where*
$$\kappa_t = \|\nabla^2 f(w_0; z_t)\|_2 + \frac{\rho}{2}\|\sum_{k=1}^{t-1} \alpha_k \nabla f(w_{S,k}; z_k)\| + \frac{\rho}{2}\|\sum_{k=1}^{t-1} \alpha_k \nabla f(w_{S';k}; z_k)\|.$$

2. $\mathbb{E}_{\mathcal{A}}[\|\Delta_{t+1}\||\Delta_{t_0} = 0] \leq$
   $\{\mathbb{E}_{\mathcal{A}}[\|\Delta_t\||\Delta_{t_0} = 0][1 + (1 - \frac{1}{n})\alpha_t\psi_t]\} + \frac{2\alpha_t L}{n}.$

3. $E_{S,S'}\{E_{\mathcal{A}}[\|\Delta_T\||\Delta_{t_0} = 0]\} \leq \frac{L}{n}\left(\frac{T}{t_0}\right)^{\zeta b}$, where
   $\zeta = \tilde{O}(\min\{\beta, E_z[\|\nabla^2 f(w_0; z)\|_2] + \Delta^*_{1,\sigma^2}\}),$
   $\Delta^*_{1,\sigma^2} = \rho(b\sigma + \sqrt{bE_z[f(w_0; z)] - k^*}$
   and $k^* = \inf_w E_z[f(w; z)].$

Based on the above lemma, we can prove an upper bound of on-average stability with uniform sampling SGD using the same technique as for Theorem 5.

**Theorem 6.** *(Data-dependent version of Theorem 5) Assume $f$ is $\beta$-smooth, $L$-Lipschitz, and has a $\rho$-Lipschitz Hessian. Let $w_t, w_{t'}$ be the outputs of SGD on twin datasets $S, S'$ respectively after $t$ iterations and let $\Delta_t = [w_t - w_{t'}]$ and $\delta_t = E_{\mathcal{A}}\|\Delta_t\|$. And let $\zeta$ follow the same definition as in Lemma 5. Running SGD on $f(w; S)$ with step size $\alpha_t = \frac{b}{t}$ where $b < 1$ satisfies*

$$\widehat{\varepsilon}_{stab} \leq \frac{16L^2 T^{\zeta b}}{\zeta n^{1+\zeta b}}. \tag{6}$$

We conclude this section with the following lower bound on the uniform stability of SGD with constant stepsize for non-convex loss functions. We show that for non-convex functions satisfying classical conditions $\beta$-smooth, we cannot avoid a pessimistic bound. Thus, in order to analyze the generalization power of SGD for deep learning loss functions from an optimization perspective, different conditions are necessary.

**Theorem 7.** *Let $w_t, w_t'$ be the outputs of SGD on twin datasets $S, S'$, and let $\Delta_t = w_t - w_t'$. There exists a non-convex, $\beta$-smooth function $f$, twin sets $S, S'$ and constants $a, \gamma$ such that the divergence of SGD after $T > n$ rounds using constant step size $\alpha = \frac{a}{0.99\gamma}$ satisfies $\varepsilon_{stab} \geq e^{aT/2}/n^2$.*

## 5 CONCLUSION AND FUTURE WORK

We first provided matching upper and lower data-independent bounds on the stability of SGD for three kinds of loss functions: convex, strongly-convex, and non-convex, essentially closing the gap in all cases. We then provided stronger data-dependent generalization bounds for both convex and non-convex loss functions by analyzing average-stability, showing that nice properties of data can both improve generalization and also reduce the need for regularization. At least two interesting open questions arise from our work: a) Can one obtain data-dependent lower bounds on average-stability that show the tightness of existing analysis? b) Can one devise properties of data-distributions or loss functions (perhaps motivated by deep learning) that imply better data-dependent stability bounds?

## Acknowledgements

We thank anonymous reviewers for their constructive feedback. Mayank Goswami would like to acknowledge support from NSF awards CRII-1755791 and CCF-1910873. Chao Chen was partially supported by grants NSF IIS-1909038 and CCF-1855760.

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
