# OpenReview forum: "Stability of SGD: Tightness Analysis and Improved Bounds"
_auai.org/UAI/2022/Conference — UAI 2022 Poster_

### Official Review · Reviewer_CeaB · 2022-04-08

**Q2(1) Originality/Novelty:** 3
**Q2(2) Significance/Impact:** 2
**Q2(3) Correctness/Technical Quality:** 3
**Q2(6) Clarity Of Writing:** 4
**Q6 Overall Score:** 5
**Q8 Confidence In Your Score:** 2

**Q1 Summary And Contributions:**

The paper studies the generlizability of Stochastic Gradient Descent (SGD) by performing data-independent tightness analysis, and providing data dependent bounds


**Q2 Assessment Of The Paper:**

More detailed information regarding each of these aspects is given below:

**Q2(5) Reproducibility:**

3: Good: Key resources (e.g., proofs, code, data) are available and key details (e.g., proofs, experimental setup) are sufficiently well-described for competent researchers to confidently reproduce the main results.

**Q3 Main Strengths:**

- The paper's topic is of high relevance to the conference and builds upon a well cited work on SGD generalization
- Presentation is well performed across the paper's sections
- Detailed results are reported, with separation to the convex and non-convex case


**Q4 Main Weakness:**

- Results are somewhat preliminary and the contribution of the paper might not be mature enough for the conference


**Q5 Detailed Comments To The Authors:**



**Q7 Justification For Your Score:**

The paper is well presented and may provide an interesting contribution to the conference's program. However the overall takeaway is marginal for a full contribution.

**Q9 Complying With Reviewing Instructions:**

1: Yes.

---

### Official Review · Reviewer_NgFt · 2022-04-13

**Q2(1) Originality/Novelty:** 3
**Q2(2) Significance/Impact:** 2
**Q2(3) Correctness/Technical Quality:** 3
**Q2(6) Clarity Of Writing:** 3
**Q6 Overall Score:** 6
**Q8 Confidence In Your Score:** 3

**Q1 Summary And Contributions:**

The paper under consideration is devoted to the stability of SGD. The authors show the tightness of results  (Hardt et al 2016) and some improvement of existing bound for the non-convex case.

**Q2 Assessment Of The Paper:**

More detailed information regarding each of these aspects is given below:

**Q2(5) Reproducibility:**

3: Good: Key resources (e.g., proofs, code, data) are available and key details (e.g., proofs, experimental setup) are sufficiently well-described for competent researchers to confidently reproduce the main results.

**Q3 Main Strengths:**

The papers provide new theoretical results on SGD and show the sharpness of existing bounds

**Q4 Main Weakness:**

It is too many theoretical results for such a short paper. It would be better to concentrate on the convex case or to make a full-length journal article.

**Q5 Detailed Comments To The Authors:**

As I mentioned above I would like to see fewer results but more details in the main part of the text

**Q7 Justification For Your Score:**

I appreciate the theoretical results presented in the current manuscript. However, I believe that presenting fewer results ( for example only a convex case), but with more details would improve the readability of the paper.

**Q9 Complying With Reviewing Instructions:**

1: Yes.

---

### Official Review · Reviewer_meUJ · 2022-04-14

**Q2(1) Originality/Novelty:** 2
**Q2(2) Significance/Impact:** 2
**Q2(3) Correctness/Technical Quality:** 3
**Q2(6) Clarity Of Writing:** 2
**Q6 Overall Score:** 4
**Q8 Confidence In Your Score:** 4

**Q1 Summary And Contributions:**

This paper shows lower tight stability bounds for general datasets and convex and strongly-convex loss functions. It also proves a lower bound for the non-convex setting and provides a corresponding upper bound.

**Q2 Assessment Of The Paper:**

More detailed information regarding each of these aspects is given below:

**Q2(5) Reproducibility:**

3: Good: Key resources (e.g., proofs, code, data) are available and key details (e.g., proofs, experimental setup) are sufficiently well-described for competent researchers to confidently reproduce the main results.

**Q3 Main Strengths:**

This paper shows lower tight stability bounds for general datasets and convex and strongly-convex loss functions. It also proves a lower bound for the non-convex setting and provides a corresponding upper bound.

Overall, the setting of this paper is based on the two references [Hardt et al., 2016] and [Kuzborskij and Lampert, 2018].

**Q4 Main Weakness:**

Since this paper extends previous framework from [Hardt et al., 2016] and [Kuzborskij and Lampert, 2018], it is not very novel. The setting of Theorem 3 is restricted, and the bounded 2nd moment assumption is not very well motivated. The new upper bound for non-convex case is not necessary better than the previous one.

Although the work [Bassily et al., 2020] is for non-smooth convex loss, it is still a reference on lower stability bounds of SGD. Hence the authors should compare their results vs this paper and highlight the differences in assumptions and complexities.



**Q5 Detailed Comments To The Authors:**

- It would be better if the Table 1 shows what theorem is corresponding to the results in this paper.
- Theorem 3 has some typos, and the assumptions are not clearly stated (bounded 2nd moment). In addition, it is not sure how the example Linear Regression satisfies that assumption.

**Q7 Justification For Your Score:**

My score is based on the (somewhat) limited novelty of this paper.

**Q9 Complying With Reviewing Instructions:**

1: Yes.

---

### Official Review · Reviewer_h1ZD · 2022-04-15

**Q2(1) Originality/Novelty:** 2
**Q2(2) Significance/Impact:** 3
**Q2(3) Correctness/Technical Quality:** 3
**Q2(6) Clarity Of Writing:** 4
**Q6 Overall Score:** 5
**Q8 Confidence In Your Score:** 4

**Q1 Summary And Contributions:**

In this paper, the authors provide (1) data-independent lower bounds on the stability of SGD for convex, strongly-convex and non-convex losses, to match the existing upper bounds; (2) improved upper bounds for non-convex losses; and (3) data-dependent generalization bounds for both convex and non-convex losses. The paper is easy to follow. However, the proofs in this work are based on some strong assumptions. Overall, the contributions are only marginally significant or novel.

**Q2 Assessment Of The Paper:**

More detailed information regarding each of these aspects is given below:

**Q2(5) Reproducibility:**

3: Good: Key resources (e.g., proofs, code, data) are available and key details (e.g., proofs, experimental setup) are sufficiently well-described for competent researchers to confidently reproduce the main results.

**Q3 Main Strengths:**

Analysing the tightness of the existing stability of SGD is interesting and practical in the machine learning community. While existing works focus mostly on deriving upper bounds for SGD, this work concentrates on analysing its lower bound.
Meanwhile, the presentation of this paper is very clear.

Some result of the paper is interesting. E.g.,

- Theorem 3 suggests that if the dataset is sampled from a ‘good’ distribution, the models can obtain an advanced generalization property.


**Q4 Main Weakness:**

The assumption that the different entry data x′i and xi are orthogonal to all other samples is very strong and unrealistic in practice. How would this assumption affect the tightness?

**Q5 Detailed Comments To The Authors:**

See Q3 and Q4

**Q7 Justification For Your Score:**

The paper study an interesting and important problem in the machine learning community: the tightness of existing algorithmic stability analysis. Through deriving lower bounds, this paper manages to (1) analyse the tightness of existing stability analysis of SGD; (2) improve the existing algorithmic stability for non-convex losses. However, some assumptions (e.g., data entries are orthogonal to all other samples) in the paper are too strong.

**Q9 Complying With Reviewing Instructions:**

1: Yes.

---

### Decision · Program_Chairs · 2022-05-15

**Decision:**

Accept (Poster)

**Comment:**

Meta Review: The authors have addressed most of the reviewers’ concerns and the reviewers have positive opinions about the paper. They agree that the paper has some contribution and are not against its publication. The authors should explain clearly the novelty of the paper compared with [Hardt et al., 2016] and [Kuzborskij and Lampert, 2018]. Moreover, all other comments and suggestions from the reviewers in their detailed reviews should be taken into account for the final version.